# Immunological Characteristics of Non-Intensive Care Hospitalized COVID-19 Patients: A Preliminary Report

**DOI:** 10.3390/jcm10040849

**Published:** 2021-02-19

**Authors:** Salvatore Corrao, Francesco Gervasi, Francesca Di Bernardo, Giuseppe Natoli, Massimo Raspanti, Nicola Catalano, Christiano Argano

**Affiliations:** 1COVID Unit, Department of Internal Medicine, National Relevance and High Specialization Hospital Trust ARNAS Civico, Di Cristina, Benfratelli, 90127 Palermo, Italy; peppenatoli@gmail.com (G.N.); massimoraspanti1991@gmail.com (M.R.); dott.catalanonicola@gmail.com (N.C.); chargano@yahoo.it (C.A.); 2Dipartimento di Promozione della Salute, Materno Infantile, Medicina Interna e Specialistica di Eccellenza “G. D’Alessandro”, PROMISE, University of Palermo, 90127 Palermo, Italy; 3Specialized Laboratory of Oncology, National Relevance and High Specialization Hospital Trust ARNAS Civico, Di Cristina, Benfratelli, 90127 Palermo, Italy; francesco.gervasi@arnascivico.it; 4Department of Microbiology and Virology, National Relevance and High Specialization Hospital Trust ARNAS Civico, Di Cristina, Benfratelli, 90127 Palermo, Italy; francesca.dibernardo@arnascivico.it

**Keywords:** SARS-CoV-2, multiparametric flow cytometry, immune system deficiency, cluster analysis

## Abstract

The outbreak of coronavirus disease 2019 (COVID-19) is posing a threat to global health. This disease has different clinical manifestations and different outcomes. The immune response to the novel 2019 coronavirus is complex and involves both innate and adaptive immunity. In this context, cell-mediated immunity plays a vital role in effective immunity against SARS-CoV-2. Significant differences have been observed when comparing severe and non-severe patients. Since these immunological characteristics have not been fully elucidated, we aimed to use cluster analysis to investigate the immune cell patterns in patients with COVID-19 who required hospitalization but not intensive care. We identified four clusters of different immunological patterns, the worst being characterized by total lymphocytes, T helper lymphocytes CD4^+^ (CD4^+^), T cytotoxic lymphocytes CD8^+^ (CD8^+^) and natural killer (NK) cells below the normal range, together with natural killer lymphocyte granzyme < 50% (NK granzyme^+^) and antibody-secreting plasma cells (ASCs) equal to 0 with fatal outcomes. In the worst group, 50% of patients died in the intensive care unit. Moreover, a negative trend was found among four groups regarding total lymphocytes, CD4^+^, CD8^+^ and B lymphocytes (*p* < 0.001, *p* < 0.005, *p* < 0.000, *p* < 0.044, respectively). This detailed analysis of immune changes may have prognostic value. It may provide a new perspective for identifying subsets of COVID-19 patients and selecting novel prospective treatment strategies. Notwithstanding these results, this is a preliminary report with a small sample size, and our data may not be generalizable. Further cohort studies with larger samples are necessary to quantify the prognostic value’s weight, according to immunological changes in COVID-19 patients, for predicting prognoses and realizing improvements in clinical conditions.

## 1. Introduction

In January 2020, the world faced the onset of the COVID-19 global pandemic, with the number of infected people increasing exponentially. This disease encompasses an asymptomatic form and different clinical manifestations, varying from fever, dry cough and dyspnea to interstitial pneumonia and acute respiratory distress syndrome (ARDS) requiring hospitalization and intensive care [1]. By 13 December 2020, 70,476,836 confirmed cases of COVID-19, including 1,599,922 deaths, were reported [2]. The immune system’s behavior plays a crucial role against SARS-CoV-2, with significant differences among severe, mild, paucisymptomatic and asymptomatic subjects [3]. An effective immune response against SARS-CoV-2 depends on innate and adaptive immunity and cytokine release [4]. A specific adaptive immune response is necessary to eliminate the virus and avoid disease progression to severe stages in the early phases. Later, IL-1β1 and IL-6—central pro-inflammatory molecules—and other cytokines and chemokines, such as TNF-α, IL-8, IL-10 and IL-2R, cause the cytokine release syndrome, which can lead to systemic clinical symptoms and organ damage, promoting the progression to more severe conditions [5,6,7]. According to current knowledge, T lymphocytes play a crucial role in an effective response against coronavirus disease 2019 and are mainly damaged by COVID-19 [8]. In this direction, Thevarajan and colleagues showed [9] the decreased function of CD4^+^ T cells, which may affect patients predisposed to severe disease, and the exhaustion of CD8^+^ T cells [10], which may reduce the cellular immune response to SARS-CoV-2. Further in this regard, Chen et al. found significant differences between the numbers of CD4^+^ and CD8^+^ T cells in severe and moderate cases [6], and Sun et al. showed a decrease in the counts of NK cells along with a decline of CD4^+^ and CD8^+^ in people with severe disease [11]. Another recent study showed that lower counts of T lymphocyte subsets were significantly associated with a higher risk of in-hospital death and disease severity [12]. The decrease in the number and dysfunction of CD8^+^ T cells may lead to a more severe course of illness and a reduction in virus eradication [13].

The evaluation of selected immune variables can help us to better understand the dynamics of immune system activation. It could represent a possible tool to set appropriate prognostic frameworks for disease outcomes and expected complications.

On the other hand, it is not clear whether the reported changes in immune cell responses are the direct result of COVID-19 or a predisposing factor for a severe course of the infection. Many questions about the immune response and its kinetics have not been resolved.

Against this background, our study aimed to perform a comprehensive evaluation of immune cell variables by cluster analysis to identify different immunological patterns among all 21 patients with COVID-19 who were admitted to the COVID Unit of the Department of Internal Medicine during the Sicilian region’s (Italy) first wave.

## 2. Methods

### 2.1. Data Collection

From 30 March to 15 April 2020, 21 consecutive patients with confirmed COVID-19 (positive RT-PCR SARS-CoV-2 and suggestive radiological signs, such as ground-glass opacities with bilateral, peripheral and basal distribution, crazy paving appearance and bronchovascular thickening in the lesions) who were admitted during the Sicilian region’s first wave to the COVID Unit of the Department of Internal Medicine of the National Relevance and High Specialization Hospital Trust ARNAS Civico, Di Cristina, Benfratelli, Italy, were enrolled in this longitudinal study. The patients were all those admitted to the Department of Internal Medicine. Written informed consent was always obtained from all patients upon admission. The Ethics committee approved the study of our institution (approval number 3143). Medical history, delay from the positive nasopharyngeal test to hospitalization and clinical, biological and immunological variables were recorded.

### 2.2. Microbiology

The molecular diagnostics of SARS-CoV-2 was carried out at the Laboratory of Virology of the Department of Microbiology using the detection of single-stranded positive-sense RNA virus in rhino-pharyngeal swabs by a reverse transcription cDNA and polymerase chain reaction (RT-PCR) (Elitech Ingenius-Arrow SeGeneNimbus, Puteaux, France).

### 2.3. Flow Cytometry

A flow cytometry six-eight-color test covering 34 different subsets of immune cells from just 2 mL of human peripheral blood (EDTA-anticoagulated) was developed. The MPFC immunophenotyping test for the direct staining of whole blood samples has been optimized. This technique allows the detection of all circulating immune cells and reduces the necessary flow cytometry preparation steps. The direct staining procedure minimizes the effort and variations in sample preparation. This represents time savings, an additional prerequisite for easy clinical application, as it requires less than 20 min.

All lymphocyte populations and T cell subpopulations, T helper, T cytotoxic, natural killer, T natural killer and B cells were determined in percentages and absolute counts. For this purpose, to allow the detailed immunophenotyping of blood, a panel with monoclonal antibodies was designed: anti-CD45, anti-CD3, anti-CD5, anti-CD8, anti-CD16, anti-CD24, anti-CD25, anti-CD27, anti-CD38, anti-CD56, anti-CD20, anti-CD45RA, anti-CD183, anti-CD196, anti-CD197, anti-TCR and anti-TCR, covering all major types of immune cells, such as T and B cells, NK cells, monocytes, neutrophils, eosinophils and plasma cells secreting antibodies (CD38^++^CD27^++^CD19^+/−^) of the IgG class (Figure 1). The panel was also used for phenotyping the T naïve/memory and T-helper 1, T-helper 2 and T-helper 17 subsets.

All monoclonal antibodies were purchased from Beckman-Coulter (Miami, FL, USA). Whole blood samples were incubated with monoclonal antibodies for 15 min at room temperature and lysed with ammonium chloride for 20 min at 4 °C by a lyse-no wash method. At least 25,000 total events were acquired, excluding doublets and debris, on a Navios^TM^ Beckman-Coulter flow cytometer (Miami, FL, USA).

The analysis of the acquired samples was carried out by the Beckman-Coulter software Kaluza Analysis 2.1 (Miami, FL, USA), with gate SS/CD45 used for the determination of lymphocyte populations and subpopulations and double gate CD19/SS and CD38/SS used for the assay of total and secreting plasma cells [14,15,16,17].

### 2.4. Statistical Analysis

Quantitative variables are summarized as medians (interquartile range: 1st–3rd) and categorical variables are reported as percentages. The whisker-plot method was used to graphically represent groups of numeric data through their quartiles and ranges. Cuzick’s non-parametric test for trend across the four groups was performed.

To detect different patterns of immune responses sharing common characteristics in the absence of an a priori hypothesis, a cluster analysis was performed. Despite this being a preliminary report, our sample fits within Forman’s rule [18], which suggests that the minimum sample size include no fewer than 2K cases (K = number of variables), and the efficacy of clustering confirmed the robustness of our analysis. For this study, Ward’s method [19] was performed as an agglomerative hierarchical clustering procedure. Unlike other methods, instead of measuring the distance directly, it analyzes the variance of clusters and represents the most suitable method for quantitative variables.

In this way, groups are formed so that the pooled within-group sum of squares is minimized. In other words, in each step, two clusters are merged together, which will result in the smallest increase in the pooled within-group sum of squares. The small sample size did not affect the discriminant capacity of the cluster analysis.

Dendrograms were output. Stata Statistical Software 2016, release 14.1 (StataCorp, College Station, TX, USA), was used for database management and analyses.

## 3. Results

During the recruitment period, 21 consecutive in-patients were investigated. Among them, 47.6% were male with a median age of 75 years old. Table 1 shows the clinical characteristics of the population.

Because of severe hypoxia, 14.3% needed intensive care hospitalization. Common clinical manifestations included fever, asthenia, dry cough, myalgia and/or arthralgia and dyspnea. The mean length of stay was 36 days. The delay from the first positive swab to hospitalization was four days (2–10). Table 2 shows the comorbidity distribution in hospitalized COVID-19 patients.

The MPFC immunophenotyping analysis (Table 3) showed that in 76.2% of patients, total lymphocytes were lower than 1200 cells/μL, with a median total number of lymphocytes equal to 620 cells/μL (380–1080).

In 61.9% of people, CD4^+^ cells were lower than 500 cells/μL, with median total CD4^+^ equal to 400 cells/μL (260–630). In one-third of patients, CD8^+^ was lower than 200 cells/μL, and in nearly one-third, natural killer cells were lower than 100 cells/μL.

In 42.9% of investigated subjects, T natural killer lymphocytes were lower than 100 cells/μL.

Interestingly, 28.6% of patients had CD8^+^ granzyme < 50% and 0 antibody-secreting plasma cells. NK lymphocyte granzyme < 50% was found in 52.4% of subjects.

The cluster analysis showed at least four main groups characterized by similar immunological frameworks (Figure 2).

The first group (median age 66 years old; range 48–83) encompassed seven patients whose CD8^+^, NK and total B lymphocytes CD 20^+^ (CD 20^+^) were within the normal range. In all patients in this group, NK lymphocyte granzyme was below 50%, while total plasma cells were within the normal range in six out of seven patients. The second group (median age 71 years old; range 56–86) included seven patients. In six out of the seven subjects, total lymphocytes were below the normal range, and CD8, NK and CD 20^+^ lymphocytes were within the normal range. All patients had T cytotoxic lymphocyte granzyme >50%, as well as NK granzyme >50%. Total plasma cells were above the normal range, and among them, hyperexpression of antibody-secreting plasma cells was found in three subjects. The third group (median age 80 years old; range 55–105) comprised three patients with total lymphocytes and CD8^+^ below the normal range. Total B lymphocytes were within the normal range, as was NK granzyme. The fourth group (median age 76 years old range; 56–86) encompassed four patients characterized by total lymphocytes below the normal range along with CD4^+^ T cells, CD8^+^, NK and NK lymphocyte granzyme < 50%. ASCs were equal to 0. The analysis of MPFC immunophenotyping of this last group showed a total number of CD3^+^ equal to (260, 130, 500, 280, respectively), CD4^+^ (110, 110, 390, 190, respectively), CD8^+^ (120, 20, 110, 150, respectively) and B lymphocytes (240, 60, 2, 30, respectively). Two out of four patients died.

After this first analysis, we wanted to test every single immunological variable to see whether there was a significant positive or negative trend between the four groups. Figure 3A shows a significant negative trend among the four groups regarding total lymphocytes, T helper lymphocytes, T cytotoxic lymphocytes CD8^+^ and B lymphocytes (*p* < 0.001, *p* < 0.005, *p* < 0.000, *p* < 0.044, respectively). On the contrary, Figure 3B shows that natural killer lymphocytes (*p* = 0.131), natural killer lymphocyte granzyme^+^ (*p* = 0.521), T cytotoxic lymphocyte granzyme^+^ (*p* = 0.760), T natural killer lymphocytes (*p* = 0.131), total plasma cells (*p* = 0.822) and Ab-secreting plasma cells (*p* = 0.497) did not have a significant trend among the four groups, nor did the number of days to negative nasopharyngeal swab (*p* = 0.241). In particular, the first group seemed to conform to the insignificant trend most of the time.

## 4. Discussion

Our analysis’s distinctive characteristic was the identification of different patterns of immunological features using cluster analysis in a group of non-intensive care hospitalized patients with COVID-19. We considered the small group a non-limiting factor since they were the whole sample of in-patients admitted in the first wave. However, given the small sample size, our data may not be generalizable. On the other hand, the cluster analysis tries to go beyond simple pairs of immune responses and to consider how immunopathological variables tend to occur in conjunction. Its conclusions are not influenced by sample size. In this way, we obtain a complete picture of the immunological population’s distribution and identify where a specific subset appears in the patterns. The crucial finding is that two out of four patterns were characterized by the lack of an adequate qualitative and quantitative immune response. One pattern is characterized by total lymphocytes and CD8^+^ below the normal range and total B lymphocytes and NK granzyme within the normal range, resulting in a partially compromised adaptive immune response. The other pattern is characterized by total lymphocytes, CD4^+^, CD8^+^, NK and NK lymphocyte granzyme <50% below the normal range, along with antibody-secreting plasma cells equal to 0, resulting in the total failure of the innate and adaptive immune response. Different studies have shown that in patients with severe COVID-19, total lymphocytes and particularly CD4^+^, CD8^+^ T cells and B cells were significantly lower than those in patients with the mild form.

CD8^+^ T cells have been indicated as an independent predictor for COVID-19 severity and treatment efficacy [8,20,21]. According to Liu and colleagues, decreased CD8^+^ was directly correlated with pulmonary involvement and pneumonia [22]. CD8^+^ can help eliminate SARS-CoV-2 by producing many biologically active molecules, such as perforin, granzyme and interferon. Therefore, the declining number of CD8+ and their dysfunction could significantly contribute to the worst course of disease and the loss of control over virus production and shedding. The exhaustion of immune cells measured by the increased expression of NKG2A (a C-type lectin receptor with inhibitory effects) on NK and CD8^+^ could contribute to an insufficient immune response [23]. In NK cells, NKG2A leads to the decreased expression of TNFα, IL-2 and IFNγ and reduced granzyme B levels. [20] Two cytokines, IL-6 and IL-10, highly present in SARS-CoV-2 infections [13], can reduce NK cell cytotoxicity [24]. In particular, IL-6 directly reduces the expression of perforin and granzyme B. In patients with COVID-19 admitted to the intensive care unit, an inverse correlation between serum levels of IL-6 and NK cells’ frequency of expressing granzyme A was found [25]. Moreover, Diao et al. showed that another exhaustion marker, PD-1 (programmed cell death 1), was significantly higher in the T cells of subjects with COVID-19, determining a loss of control over virus replication and a progression to more severe disease [13]. Wu et al. [26] reported a decline in T cells and CD4^+^ in subjects with an ineffective and unbalanced immune response. A lower number of CD4^+^ may predict a longer duration of viral RNA in the stools of patients with COVID-19 [27]. Xu et al. found that a significant decrease in the T lymphocyte subset and B cell numbers positively correlated with in-hospital death and illness severity [12]. Recent studies have shown that patients admitted to non-intensive care units with total counts of T cells, CD4^+^ cells and CD8^+^ cells lower than 800, 400 and 300/μL, respectively, need much more attention due to the high risk of further deterioration, even in the absence of severe symptoms (13). In addition, lower values of T lymphocytes and their subsets (total CD3^+^ <200/μL, CD4^+^ <100/μL and CD8^+^ <100/μL) are significantly associated with a higher risk of hospital death due to COVID-19. According to Xu et al., the warning values of lymphocytes, CD3^+^, CD4^+^, CD8^+^ and CD19^+^ cells are 559/μL, 235/μL, 104/μL, 85/μL and 82/μL, respectively, and predict in-hospital death [12]. Therefore, a lower number of CD8^+^ T cells, along with decreased counts of CD3^+^, CD4^+^, CD20^+^ and NK cells, could significantly contribute to the worst outcomes of COVID-19 [22,28]. Our data confirm the analysis mentioned above regarding the group of patients with a mild form of COVID-19 characterized by an immune deficiency that involves both total lymphocytes and subsets. This immune gap is particularly evident when we analyze the significant differences among the four groups concerning total lymphocytes, CD8, CD4 and CD20. These findings lead to the identification of a pattern of immunological features characterized by innate and acquired immune deficiency and a lack of antibody-secreting plasma cells related to in-hospital death and disease severity. Another point worth mentioning is the identification of a pattern of immunological variables characterized by total plasma cells exceeding regular levels with the hyperexpression of antibody-secreting plasma cells.

Our data agree with Varnaitė et al. [29]. They found that in COVID-19 patients, there was a SARS-CoV-2–specific B cell response indicated by the hyperexpression of antibody-secreting plasma cells 19 days after COVID-19 symptom debut, highlighting an active B cell response against the SARS-CoV-2 virus. In addition, our findings are similar to another study by Lee et al. [30], who pointed out that in acute respiratory syncytial virus infection, antibody-secreting cell expansion could be detected 22–45 days after the onset of symptoms due to the longer acute respiratory syncytial virus shedding time in the respiratory tract. Therefore, the kinetics of the ASC response might depend on the persistence of the pathogen. Interestingly, in contrast to Theravaian and colleagues [3], who found that antibody-secreting cells appeared on day seven, peaked on day eight along with T helper cells and reached peak levels on day 20 during recovery, our data show that in some COVID-19 patients, there is a total lack of an immune response, even after more than one month from the first positive swab test. These pieces of evidence raise questions about the production of a protective T cell memory following SARS-CoV-2 infection. Grifoni et al. identified specific memory CD4^+^ and CD8^*^ T cells in ~70% and 100% of COVID-19 convalescent patients [31]. SARS-CoV-2 spike glycoprotein (S)-reactive CD4^+^ cells were detected in 83% of patients with COVID-19 [32], as well as in 35% of healthy subjects demonstrating the presence of S-cross-reactive T cells, probably generated during previous infections with endemic coronaviruses. In addition, memory T cells were also detected for nucleoprotein and membrane protein. Recent studies have shown that T cell responses’ breadth and magnitude were significantly higher in severe cases compared to mild cases [33]. However, the duration of SARS-CoV-2 neutralizing antibodies remains unclear. According to Li et al., SARS-CoV-2 neutralizing antibodies were detected one-year post-infection in convalescent patients with severe acute respiratory syndrome caused by coronavirus [34]. Recent studies have shown that a cellular immune response against SARS-CoV-2 persists for about six months after mild or asymptomatic disease, although the immune response was 50% stronger in subjects affected by the symptomatic disease [35]. Other studies will be necessary to define the durability of T cell memory.

In conclusion, our preliminary findings are in agreement with literature data. Nevertheless, to our knowledge, this is the first study that identifies at least four different clusters, not a priori, in contrast to previous investigations that focused on variations of individual cell lines. Obviously, our data are from a small sample size, and larger investigations should confirm our findings. Further studies could investigate these different acquired and innate immune failure patterns to carefully assess and follow this condition’s course and its outcomes.

## Figures and Tables

**Figure 1 jcm-10-00849-f001:**
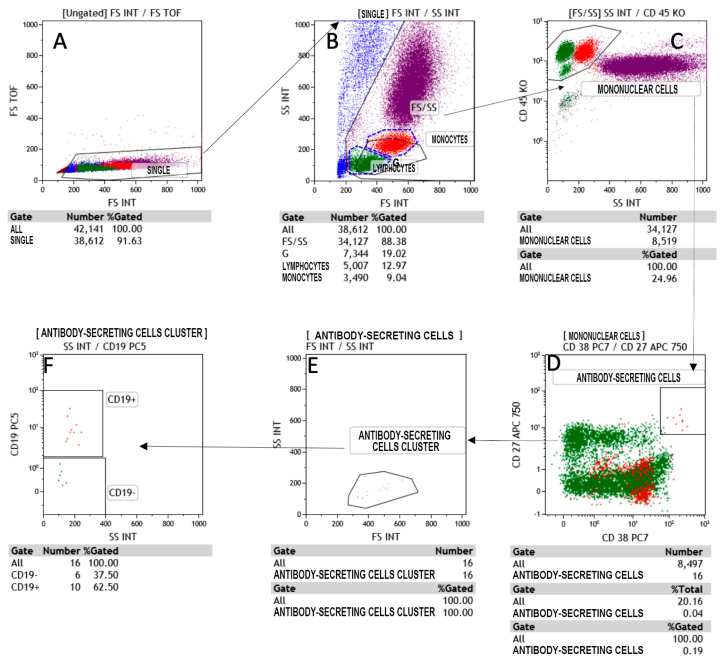
Flow cytometry gating strategy to identify the antibody-secreting cells (ASC) in Covid-19 patients. (**A**) The dot plot A shows the region SINGLETS; (**B**) dot plot B gated on SINGLETS region display physical characteristic FS vs SS in which we are drawn lymphocytes and monocytes region, (**C**) in dot plot C gated by FS/SS region the intersection between CD45 vs SS identify the Mononuclear cells gate; (**D**) the antibody secreting cells (ASC) shown in dot plot D, are gated by mononuclear cells. Finally; (**E**,**F**) in E and F the cell cluster and the CD19 expression are respectively characterized.

**Figure 2 jcm-10-00849-f002:**
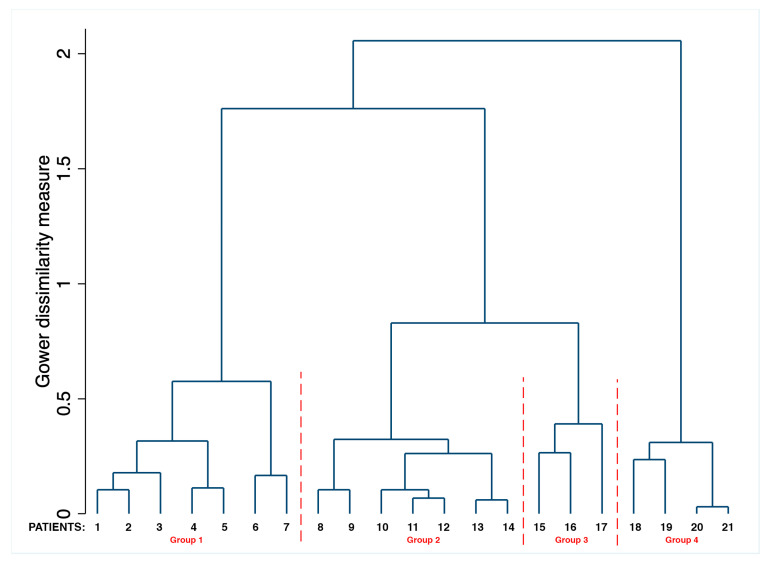
Dendrogram resulting from cluster analysis testing for the distribution and aggregation of immune response variables in hospitalized COVID-19 patients.

**Figure 3 jcm-10-00849-f003:**
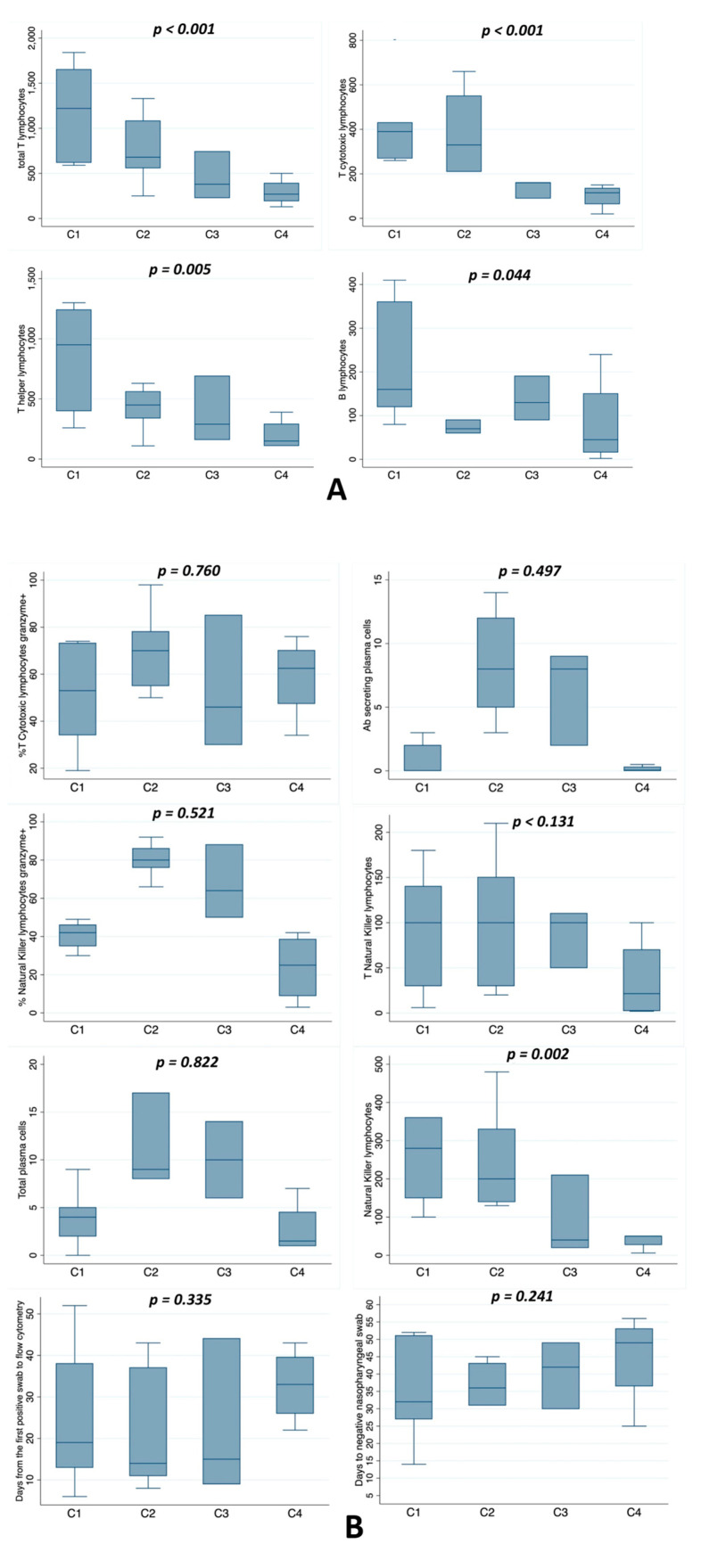
(**A**) Whisker-plot comparison of the four groups according to flow cytometry with a statistically significant trend from the first to the fourth group. The *p*-value is referenced to indicate the presence or absence of a significant trend among the four groups. (**B**) Whisker-plot comparison of the four groups according to variables without a statistically significant trend from the first to the fourth group. The *p*-value is referenced to indicate the presence or absence of a significant trend among the four groups.

**Table 1 jcm-10-00849-t001:** Clinical characteristics of the hospitalized COVID-19 patients.

Variables	
*n*	21
Age ^§^	75 (61–81)
Men (%)	47.6
Low flow oxygen (%)	66.7
High flow oxygen (%)	14.3
Intensive care unit hospitalization (%)	14.3
Days from the first positive swab to flow cytometry ^§^	19 (13–37)
Days from the first positive swab to hospitalization ^§^	4 (2–10)
Days from first to last swab ^§^	42 (31–48)
Last positive swab (%)	28.6
Deaths (%)	9.5
Hospital stay ^§^	36 (30–43)
White blood cells (×10^3^/μL) ^§^	5.1 (4.0–7.8)
Fever (%)	57.1
Asthenia (%)	40.0
Dry cough (%)	26.7
Myalgia and or arthralgia (%)	26.7
Dyspnea (%)	20.0
Chest Pain (%)	13.3
Anorexia (%)	6.7
Nausea (%)	6.7
Diarrhea (%)	6.7

^§^ Data are reported as medians (Q1–Q3).

**Table 2 jcm-10-00849-t002:** Comorbidity distribution in hospitalized COVID-19 patients.

Variables	
Hypertension (%)	72.2
Cerebrovascular disease (%)	38.9
COPD (%)	33.3
Atrial fibrillation (%)	27.8
Chronic renal failure (%)	27.8
Heart failure (%)	27.8
Dyslipidemia (%)	22.2
Ischemic heart disease (%)	22.2
Obesity (%)	16.7
Dementia (%)	16.7
Diabetes (%)	16.7
Metabolic syndrome (%)	11.1

**Table 3 jcm-10-00849-t003:** Immune cell layout of hospitalized COVID-19 patients.

Variables	
Total T lymphocytes ^§^	620 (380–1080)
Total T lymphocytes <1200 (%)	76.2
1200 ≤ total T lymphocytes ≤ 1400 (%)	9.5
Total T lymphocytes >1400 (%)	14.3
T helper lymphocytes CD4^+ §^	400 (260–630)
T helper lymphocytes CD4^+^ <500 (%)	61.9
500 ≤ T helper lymphocytes CD4^+^ ≤ 2000 (%)	38.1
T helper lymphocytes CD4^+^ >2000 (%)	0.0
T cytotoxic lymphocytes CD8^+ §^	270 (160–410)
T cytotoxic lymphocytes CD8^+^ <200 (%)	33.3
200 ≤ T cytotoxic lymphocytes CD8^+^ ≤ 1200 (%)	66.7
T cytotoxic lymphocytes CD8^+^ >1200 (%)	0.0
Natural killer lymphocytes ^§^	150 (50–300)
Natural killer lymphocytes <100 (%)	28.6
200 ≤ Natural killer lymphocytes ≤ 1200 (%)	71.4
Natural killer lymphocytes >1200 (%)	0.0
B lymphocytes CD20^+ §^	90 (70–160)
B lymphocytes CD20^+^ <60 (%)	9.5
60 ≤ B lymphocytes CD20^+^ ≤ 800 (%)	90.5
B lymphocytes >800 (%)	0.0
T Natural killer lymphocytes ^§^	100 (30–110)
T Natural killer lymphocytes <100 (%)	42.9
100 ≤ T Natural killer lymphocytes ≤ 500 (%)	57.1
T Natural killer lymphocytes >500 (%)	0.0
% T cytotoxic lymphocytes CD8^+^ granzyme^+ §^	62 (46–74)
% T cytotoxic lymphocytes CD8^+^ granzyme^+^ <50% (%)	28.6
% T cytotoxic lymphocytes CD8^+^ granzyme^+^ ≥ 50% (%)	71.4
% Natural killer lymphocytes granzyme^+ §^	49 (38–77)
% Natural killer lymphocytes granzyme^+^ <50% (%)	52.4
% Natural killer lymphocytes granzyme^+^ ≥ 50% (%)	47.6
Total plasma cells ^§^	7 (2–9)
Total plasma cells <1 (%)	14.3
1 ≤ Total plasma cells ≤ 11 (%)	66.7
Total plasma cells >11 (%)	19.0
% Total plasma cells ^§^	7.9 (2.5–15)
% Total plasma cells <0.7 (%)	4.8
0.7 ≤ % Total plasma cells ≤ 4.8 (%)	42.8
% Total plasma cells >4.8 (%)	52.4
Ab-secreting plasma cells ^§^	2 (0.5–8)
Ab-secreting plasma cells <1 (%)	28.6
1 ≤ Ab-secreting plasma cells ≤ 5 (%)	38.1
Ab-secreting plasma cells >5 (%)	33.3

^§^ Data are reported as medians (Q1–Q3).

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
