# Peer review of "Immunological Characteristics of Non-Intensive Care Hospitalized COVID-19 Patients: A Preliminary Report"

_jcm, 2021, doi:10.3390/jcm10040849_

Round 1

Reviewer 1 Report

Review of the paper:

Immunological characteristics of non-intensive care hospitalized COVID-19 patients: preliminary report.

Minor details:

1) The word “Methods” is repeated in lines 62 and 70

2) Re-phrase Lines 107 and 108 because it is hard to understand.

“For the purpose of this study the Ward’s method [15], an agglomerative hierarchical cluster analysis, was performed. ”

3) Line 147: “Six out seven subjects had total lymphocytes below the normal range, T cytotoxic lymphoc…”

The proper way is six out of seven…

4) Figure 2 The groups are indicated as Gruppo. Translate to English

5) If those acronyms in figure 2 are the initials of the patient’s names, they should be removed to keep the patient’s privacy.

6) Line 157 Two out four patients died, the correct grammar is Two out of four patients died.

Major details:

Discussion:

The discussion is very daring compared to the results showed. Some conclusions cannot be achieved with the results showed, for instance in line 201 the following is indicated:

“Intriguing, we observed a significant expansion of antibody secreting plasma cells 24.5 days after the first positive swab.”

The authors did not mention time component until this point, and this is not supported by any result.

Figure 3 is mentioned in the results but no discussion is made, regarding differences and similarities observed in the different boxplots.

Recommendations:

1) I recommend that the article be reviewed by a native English speaker, since there are several paragraphs with grammar mistakes.

2) I strongly recommend contrasting these results with patient’s genders, since it is not clear if the observed differences are associated with this feature.

3) Also, these differences could be due to age strata, so I strongly recommend including in figure 2, besides the gender, the age of the patients, to determine if the differences are really due to immune responses and not due to other features.

4) Finally I recommend that the samples need to be synchronized in time because it is clear that some patients are exhibiting acquired immune response while others are still in the innate phase of the immune response

5) I recommend to perform an analysis establishing 3 groups instead of 4, merging groups 2 and 3 into one group. This mainly because the distance between these 2 groups is relatively small compared with the other groups.  It is important to evaluate the models through grouping metrics, so the authors can determine the most adequate model (3 groups or 4 groups), because so far, the analysis is too preliminary and in my opinion, it is not sufficient enough to be considered for publication.

Author Response

Thank you for your helpful comments to improve our manuscript:

  • According to your suggestion we deleted the repeated word “Method” page 2, line 72.
  • According to your suggestion we re-phrased “In order to analyze different patterns of immune response, without any a priori hypothesis, a cluster analysis was performed. For the purpose of this study the Ward’s method (15), an agglomerative hierarchical cluster analysis, was performed” page 5, lines 118-124.
  • According to your suggestion we have edited “six out of seven” page 8, line 167.
  • According to your suggestion we have edited” two out of four patients” page 8, lines 177.
  • According to your suggestion we have translated “Gruppo” to “Group” in figure 2 page 8, line 160.
  • According to your suggestion we have removed the initial of the patient’s name page 8, line 159.
  • According to your suggestion we have deleted the statement “Intriguing, we observed a significant expansion of antibody secreting plasma cells 24.5 days after the first positive swab”.
  • According to your suggestion we have better discussed the results. page 11,12 lines 215-245.
  • According to your suggestion we have edited English language.
  • According to your suggestion we have contrasted these results with patient’s gender using gender as a variable for the cluster analysis. Cluster analysis report did not change.
  • According to your suggestion we have reported the median age of single group page 8 line 164, 167,171, 173
  • To answer to your suggestion, we have reported “Days from the first positive swab to flow cytometry” in table 1.
  • According to your suggestion we have performed an analysis establishing three groups instead of four groups. However, in our opinion the third group was closer to the fourth rather than the second. So, we have preferred to leave four groups, not overturning the meaning of the cluster analysis.

Reviewer 2 Report

The manuscript “ Immunological characteristics of non-intensive care hospitalized COVID-19 patients: preliminary report” has analysed various patterns of Immunological features in non-intensive care hospitalised COVID-19 patients. There are some major and minor flaws with the current manuscript that need to be addressed.

Major

  1. The results section needs to be critical and relevant studies needs to be added, briefly discussed, and concluded.

Minor

  1. The English in the manuscript is poor and requires editing.
  2. In the introduction, mechanisms of dysregulated immune responses (cytokine storm etc) and COVID infection characteristics by three vital symptoms fever over 38 °C, dyspnea, and dry coughs should be added.
  3. Please discuss the role of memory T cells in COVID-19 patients. Do you expect to observe high memory cells as reported in different studies? (doi:1016/j.cell.2020.05.015 - doi:10.1038/s41586-020-2550-z -https://doi.org/10.1038/s41586-020-2598-9
  1. Gating strategy could be added to supplementary material.

Author Response

Thank you for your helpful comments to improve our manuscript:
• According to your suggestion we have modified the result section from page 5 line 131 to page 10 line 195.
• According to your suggestion we have added and discussed new relevant studies in the discussion section page11, from line 215 to line 230.
• According to your suggestion we have edited English language.
• According to your suggestion we have explained the mechanism of dysregulated response better and we have added COVID infection characteristics and symptoms in the introduction section, page 3 from line 46 to line 52.
• According to your suggestion we have discussed the role of memory T cell page 12, from line 257 to line 269.
• If you agree with us, we would prefer to leave the gating strategy inside the manuscript.

Reviewer 3 Report

The authors of the manuscript made an attempt to assess the immune properties of patients with mild COVID-19.

Unfortunately, the authors seem to dash off a short paper very quickly. Please find below my comments:
1. All abbreviations should be explained.
2. The number of COVID-19 patients and the number of deaths should be modified as of the date of submitting the article to JCM.
3. Introduction reads like randomly "stuck" sentences together.
4. Is the immunological nomenclature used correct - CD4 or CD4 +?
5. The authors in the introduction cite single studies, unfortunately without logical sense and do not provide conclusions from these studies.
6. The aim of the study was not clearly specified. What is unique about this manuscript compared to the previous articles?
7. The number of enrolled patients (n-21) is embarrassingly low.
8. I cannot find in the text the consent of the bioethics committee to proceed with the research.
9. The definition of patients should be more precise. What does "radiological signs" mean? Moreover, PCR and not pcr.
10. What was the optimization of flow cytometry? In general, this analytical method has been poorly characterized.
11. Figures are of poor quality.
12. The English language is used badly. Apart from linguistic errors, the authors also make mistakes that probably result from sloppiness, eg Miami, Flo should be Miami, FL, USA. You can find many such examples in the text.
13. After the description of flow cytometry, the authors carelessly describe the PCR - there is no logical chronology here.
14. Ward's method - I am not convinced that it can be used for 21 patients.
15. The patients are very poorly characterized and, in fact, no clinical and demographic data are available.
16. The results are confused and difficult to analyze as they are.
17. The discussion is trivial.

Author Response

Thank you for your helpful comments to improve our manuscript:
• According to your suggestion we have explained all the abbreviations.
• According to your suggestion the number of COVID-19 patients and the number of deaths were modified as the date of submitting the article to JCM page 3, line 44,45.
• According to your suggestion we have revised the introduction.
• According to your suggestion we have edited the immunological nomenclature.
• According to your suggestion we have clearly specified the aim of the study page 3, lines 68-70.
• We agree with your observation about the low number of patients investigated but this is the whole sample available in the first wave. Notwithstanding, the sample size did not affect the cluster analysis.
• According to your suggestion we have reported the consent of the bioethics committee page 3, lines 79,80.
• According to your suggestion we have explained the significance of “radiological signs” page 3, line 74,75.
2
• We disagree with your comment about the flow cytometry because we characterized this method step by step.
• According to your suggestion we have improved the quality of figures.
• According to your suggestion we have edited linguistic errors.
• According to your suggestion we have modified the chronology of PCR description page 4, lines 83-86.
• According to your suggestion we have explained better the reason why we have used the ward’s method page 5, lines 118-124
• According to your suggestion we have characterized patients better see page 5, line 134 table 1, and page 6, line142 table 2.
• According to your suggestion we have modified the result section from page 5 line 131 to page 10 line 195.
• According to your suggestion we have modified the discussion section from page 11 line 201 to page 12 line 274.

Round 2

Reviewer 1 Report

Second Review:

Immunological characteristics of non-intensive care hospitalized COVID-19 patients: preliminary report.

The article is much better now, especially the discussion.

Minor details:

I recommend including an "a" in the title as follows

Immunological characteristics of non-intensive care hospitalized COVID-19 patients: a preliminary report.

Major details:

1) It is necessary to report and explain in the methods section, the statistical method used to determine the differences between boxes in the boxplots. In other words, what is the test for a significant trend among the groups?

Please be more specific in the methodologies used because some of the differences in the boxplots, especially in figure 3b seem to be statistically significant.

Author Response

Thank you for your helpful comments to improve our manuscript:

  • According to your suggestion we have included an “a” in the title
  • According to your suggestion we have reported the test for a significant trend among the four groups page 5, line 117.
  • Looking at the medians and the interquartile distances among the four groups, statistical significance seems consistent with the conclusions. We have better specified in the legends page 9, line 189, page 10, line 194.

Reviewer 2 Report

The authors have addressed the questions and properly improved the manuscript.

Author Response

Thank you for your helpful comments

Reviewer 3 Report

Despite some significant improvements, the article adds nothing new to understanding the pathophysiology of COVID-19. The conclusions proposed by the authors are not correct for the presented number of patients.

Author Response

Thank you for your helpful comments to improve our manuscript:

  • According to your suggestion we have edited English language further.
  • Thank you for your opinion, but these 21 cases represent an example of what occurs in the real life. Actually, we cannot exactly quantify the phenomenon even if the cluster analysis helps to summarize and analyze the data but in light of the evidence, we do not believe that our hypothesis has to be underestimated. According to our opinion, it is mandatory acknowledge our findings.